# The Pre-BRA (pre-pectoral Breast Reconstruction EvAluation) feasibility study: protocol for a mixed-methods IDEAL 2a/2b prospective cohort study to determine the safety and effectiveness of prepectoral implant-based breast reconstruction

Kate Louise Harvey,[1] Nicola Mills,[1] Paul White,[2] Christopher Holcombe,[3] Shelley Potter ![ORCID] ,[1,4] on behalf of The Pre-BRA Feasibility Study Steering Group

For numbered affiliations see end of article.

**Correspondence to**
Kate Louise Harvey;
kate.harvey@bristol.ac.uk

## ABSTRACT

**Introduction** Implant-based breast reconstruction is the most commonly performed reconstructive technique worldwide. Subpectoral reconstruction with mesh is the current standard of care but new prepectoral techniques have recently been introduced. Prepectoral breast reconstruction (PPBR) may improve outcomes for patients but robust evaluation is required. Randomised clinical trials (RCTs) are ideally needed but the short-term safety of PPBR is yet to be established; the technique and its indications are evolving and it has yet to be adopted by a sufficient number of surgeons for an RCT to be feasible. The Pre-BRA study aims to determine the feasibility of using mixed-methods within an IDEAL 2a/2b (IDEAL, Idea-Development-Exploration-Assessment-Long-term) study to explore the short-term safety of PPBR and determine when the technique is sufficiently stable for evaluation in a pragmatic RCT.

**Methods and analysis** Pre-BRA is an IDEAL stage 2a/2b prospective multicentre cohort study with embedded qualitative research.

Consecutive patients electing to undergo immediate PPBR at participating centres will be invited to participate. Demographic, operative, oncology and complication data will be collected and patient-reported outcomes will be assessed at baseline, 3 and 18 months postoperatively. The primary safety endpoint will be implant loss at 3 months.

Surgeons performing PPBR will be asked to complete questionnaires regarding their practice and report any modifications made to the procedure or learning arising from complications via free-text response fields on electronic case-report forms. Semistructured will explore surgeons' experiences in detail to identify emerging best practice. This will be fed back to participating surgeons to promote shared learning.

The Pre-BRA study will aim to recruit 341 patients from 30 to 40 UK centres over a 12-month period. Recruitment will commence Spring 2019.

## Strengths and limitations of this study

► Multicentre prospective cohort study which will generate high-quality data to support the safety of a novel approach to implant-based breast reconstruction prior to undertaking a definitive randomised controlled trial (RCT).

► The study will assess the feasibility of using an innovative mixed-methods approach to promote shared surgical learning and establish when a novel surgical intervention is sufficiently stable for evaluation in the context of an RCT.

► The Pre-BRA (BRA, breast reconstruction evaluation) study is a single-arm study without an in-built comparison group but the results will be compared safety data from recently published iBRA study and national quality standards.

► The feasibility of using mixed-methods to share learning and determine when the procedure has stabilised will depend on surgeon engagement, and the feasibility of this approach is yet to be explored.

**Ethics and dissemination** The study has full ethical approval from OXFORD-B South Central Committee Ref:19/SC/0129. Results will be presented at national and international meetings and published in peer-reviewed journals.

**Trial registration number** ISRCTN11898000; Pre-results.

## INTRODUCTION

Up to 40%[1] of the 55 000[2] women diagnosed with breast cancer each year in the UK will require a mastectomy and of these, approximately one in four will elect to have an immediate breast reconstruction[3] to improve their quality of life.[4] Implant-based breast

reconstruction (IBBR) is the most commonly performed technique worldwide[5][6] and subpectoral implant-based breast reconstruction with biological or synthetic mesh has become the standard of care despite the lack of high-quality data to support the safety or effectiveness of the technique.[7][8]

Recently, however, mesh-based techniques have evolved further with early studies[9] suggesting good results when fixed-volume implants are completely covered in a biological[10–12] or synthetic mesh[13–16] and placed on top of the muscle in a prepectoral position. This 'muscle-sparing' technique may result in less postoperative pain, more natural results and may prevent 'implant animation', the upwards movement of the implant seen when the pectoral muscle contracts.[17]

While early results[18] of this prepectoral technique are promising in the hands of a few expert surgeons,[10 11 13–15 19 20] historically subcutaneous implant placement without mesh, was abandoned by the reconstructive community due to high complication rates.[21–24] To date, few well-designed prospective studies have directly compared prepectoral and subpectoral IBBR techniques,[25] and evidence to support the proposed benefits of prepectoral implant placement is inconsistent.[17 26 27] There is therefore a need to robustly evaluate prepectoral implant reconstruction (PPBR) before it becomes standard practice.[18] Randomised clinical trials (RCTs) provide the best evidence, but the safety of prepectoral breast reconstruction has yet to be established; the technique and its indications are still evolving, and it has not yet been adopted by a sufficient number of surgeons in the UK for a randomised trial to be feasible. Preliminary work is therefore needed before definitive evaluation can be considered.

The IDEAL (Idea-Development-Exploration-Assessment-Long-term Study) Framework provides recommendations for the evaluation of a surgical innovation from first-in-man to long-term study.[28 29] Phases 2a (Development)/2b (Exploration) focus on feasibility work for a definitive future trial including establishing the risks and benefits of the technique; the stability of the intervention, its indications and the adoption of the procedure by a wider group of surgeons from the initial innovators.[30] While the scope and intention of IDEAL is clearly defined, the system is not well operationalised, and methods for the optimal conduct of early-phase surgical studies are unclear. Embedding qualitative methods within an IDEAL study may provide a novel way of enhancing shared learning within the surgical community to reduce learning curve effects and allow a new procedure to be introduced more safely and efficiently. Monitoring and sharing modifications may also provide an effective way of determining when the procedure is sufficiently stable for formal evaluation in the context of a definitive RCT.

## METHODS AND ANALYSIS
### Primary aim
The aim of Pre-BRA is to determine the feasibility of using mixed-methods within an IDEAL 2a/2b study to explore the short-term safety and effectiveness of PPBR and determine when it is sufficiently stable for formal evaluation in a definitive RCT.

### Specific objectives
1. To establish the short-term safety (implant loss; infection; reoperation and readmission at 3 months) and effectiveness (patient-reported outcomes using BREAST-Q at 3 and 18 months) of PPBR compared with published national standards from the National Mastectomy and Breast Reconstruction Audit[31] and iBRA[7] studies.
2. To explore the feasibility of using a novel mixed-methods approach in an IDEAL 2a/2b study to promote shared learning and identify intervention stability including the acceptability of the methodology to surgeons and the most effective methods for providing feedback.
3. To inform the feasibility, design and conduct of a future RCT including the numbers of patients at participating centres undergoing IBBR who are eligible for PPBR; types of meshes used and approach to concomitant interventions.
4. To build capacity to deliver a future RCT by establishing a network of surgeons engaged in the need for evaluation who are able to perform PPBR to a high standard and who will participate in a future study.

### Study design
The Pre-BRA study is a multicentre, IDEAL phase 2a/2b prospective observational cohort study with embedded qualitative methods.

### Patient and public involvement
This project is an extension of the iBRA study[7] which was designed with patient coapplicants. A patient group will be established to review and comment on the results. This patient group will input on the design of the future RCT, in particular, selection of the most appropriate primary outcome measure and timing of assessment.

### Setting
All UK breast and plastic surgical centres currently performing PPBR using any technique will be invited to participate through the UK professional associations (Association of Breast Surgery (ABS), British Association of Plastic Reconstructive and Aesthetic Surgeons (BAPRAS)).

All surgeons currently performing PPBR will be eligible to participate in the study but those wishing to start performing the technique will be encouraged to follow the ABS/BAPARS guidelines regarding mentorship and training[32] prior to entering patients.

### Participants
#### Inclusion criteria
Consecutive women aged 16 or over who require a mastectomy for breast cancer or risk-reduction who elect to undergo IBBR are considered technically suitable

for PPBR by their surgeon and consent to undergo the procedure are eligible to participate in the study. Patients undergoing PPBR will be informed that the procedure is innovative and that outcome data are limited as part of their discussion with the surgical team.

## Exclusion criteria

Absolute and relative exclusion criteria will be used to reflect current practice and account for the variable experience of surgeons performing PPBR across the UK. Absolute exclusion criteria will be applied to all patients. Relative exclusion criteria can be applied flexibly according to the surgeon's experience with the procedure in conjunction with the patient's consent following discussion that the risk of complications may be increased in this setting.

   Absolute exclusion criteria
i.    Patients assessed as having insufficient soft tissue coverage for PPBR by the operating surgeon.
ii.   Patients having revisional or delayed breast reconstruction.
iii.  Patients unable or unwilling to provide informed consent.

   Relative exclusion criteria reflect ABS/BAPRAS guidelines for mesh-assisted breast reconstruction[32] and include the following:
i.    Current smokers or recent ex-smokers (<6 weeks).
ii.   Previous breast/chest wall radiotherapy.
iii.  Body mass index (BMI) of greater than 30.
iv.   Reconstruction with an implant of more than 600cc.
v.    Patients anticipated to require postmastectomy radiotherapy.
vi.   Poorly controlled diabetes.

## Participant identification, screening, and recruitment

All female patients aged 16 or over undergoing mastectomy for breast cancer or risk-reduction who elect to undergo immediate IBBR will be screened to inform the feasibility of a future trial comparing different approaches to IBBR.

### Participant screening

Potential participants will be identified from clinics and multidisciplinary meetings by the clinical and research teams.

   A comprehensive screening log will be maintained at participating sites of all women who elect to undergo immediate IBBR under the care of participating surgeons to determine the proportion of the following patients:
1. Patients who are considered technically suitable for prepectoral technique by the operating surgeon.
2. For those patients who are considered not suitable for PPBR—with reasons for this (eg, relative exclusion criteria: smoker; previous RT, etc, or other reason).
3. For patients considered technically suitable for PPBR, the proportion who accept the technique.
4. For those patients accepting PPBR, the proportion who consent to study participation.

### Patient recruitment

Patients who elect to undergo PPBR will be given a patient information sheet explaining the study. They will be seen at a follow-up appointment by a member of the clinical or research team and asked to sign a consent form prior to participating in the study (online supplementary appendix 1). Patients will be allocated a study ID number.

   Patients will be given a date for surgery as per unit protocol and undergo preoperative assessment. They will be required to complete baseline patient-reported outcome questionnaires (BREAST-Q) and have preoperative photographs as per unit policy prior to surgery.

## Study procedures

All patients in whom prepectoral reconstruction with or without mesh is planned will be eligible for inclusion in the study. All meshes must be licensed for use in the UK and have a CE mark. Composite prepectoral reconstructions using dermal sling and mesh or two different types of mesh will be permitted and the details of products used will be recorded. Reconstructions using full dermal sling coverage will also be included.

### Standardisation of surgical technique and procedure fidelity

All patients will undergo a skin or nipple-preserving mastectomy followed by an immediate IBBR.

   Participating surgeons will undertake the procedure as per their standard practice. Mesh choice (biological or synthetic and the product used) and implant selection (fixed-volume; adjustable implants or tissue expanders) will be as per surgeon preference. All patients will be planned for a prepectoral reconstruction prior to surgery.

   The following steps of the PPBR procedure will be considered mandatory, prohibited and discretionary according to the typology described by Blencowe et al.[33]
► *Mandatory*—insertion of a tissue expander/adjustable implant or fixed-volume implant.
► *Prohibited*—raising the pectoralis muscle.
► *Discretionary*—all other steps of the procedure.

   If the surgeon considers that prepectoral reconstruction is not possible at the time of surgery or modifies the planned procedure (eg, inserts tissue expander rather than fixed-volume implant), this will be recorded on the case report form (CRF) with reasons for the modification (eg, thin skin-flaps). Details of the alternative procedure performed (eg, subpectoral reconstruction) will be collected. Surgeons will also be encouraged to record any learning arising from the case.

   Strategies to minimise infection (eg, use of laminar flow, cavity irrigation, and glove change) will be as per local practice but participating centres will be encouraged to adhere to published best practice guidelines[32 34] and use the evidence-based Manchester checklist.[35]

   Use of drains and other concomitant interventions (eg, antibiotics, dressings) will be permitted as per local practice.

### Deviations from planned prepectoral reconstruction

As detailed above, surgeons will be asked to report the following via the CRF:

a. Patients in whom PPBR was planned but had to be abandoned completely (ie, another form of reconstruction or no reconstruction performed).
   i. Reasons why prepectoral approach was abandoned.
   ii. Procedure performed instead.
   iii. Any reflections or learning regarding the case.
b. Patients in whom the planned procedure was modified (but prepectoral reconstruction still performed).
   i. Details of the modification made (eg, insertion of tissue expander rather than fixed volume implant).
   ii. Reasons for the modification (eg, thin skin flaps).
   iii. Any reflections or learning regarding the case.

## OUTCOME MEASURES
### Safety and effectiveness outcomes

The primary outcome for the prepectoral safety study will be implant loss at 3 months defined as the removal of the implant or tissue-expander without replacement as a result of a complication. This is consistent with the definition used in the iBRA study.[7] Secondary safety and effectiveness outcomes:

i. Postoperative pain assessed by patient self-report using a visual analogue scale at 24 hours, 1 week, 2 weeks and 3 months.
ii. Complications requiring readmission and/or reoperation at 3 months.
iii. Infection defined as (i) minor—requiring treatment with oral antibiotics, (ii) major requiring admission for intravenous antibiotics and/or surgical debridement.
iv. Quality of life including satisfaction with breasts; physical well-being; psychosocial well-being and animation assessed using the validated BREAST-Q[36] questionnaire at 3 and 18 months following surgery.
v. Further surgery for complications or cosmesis at 18 months

### Feasibility outcomes

Surgeon engagement will be used to assess the feasibility of using mixed-methods to explore learning and intervention stability in the study. At study entry, the number of surgeons completing the surgeon questionnaire; the number of surgeons consenting to be contacted for interview and the number of surgeons interviewed will be compared. For modifications/learning arising from complications, the numbers of events (modifications and complications) reported will be compared with numbers of free-text responses entered; the number of surgeons consenting to interview and the actual number interviewed. The content of the free-text responses and interviews will also be assessed.

Screening logs will be reviewed to determine the relative proportions of patients electing to undergo IBBR who are (i) considered technically suitable for prepectoral reconstruction and (ii) elect to undergo surgery to determine if a full-scale pragmatic RCT is feasible.

## DATA COLLECTION
### Clinical data

Demographic, comorbidity and operative data will be collected for each participant via a standardised electronic CRF hosted on REDCap[36] (online supplementary appendix 2).

Complications and oncological data will be collected at 30 days and 3 months by clinical or case-note review according to local follow-up policies. Details of any additional surgery required for complications or cosmesis will be collected at 18 months by case note review. No additional clinic visits will be required for the study. All complications will be defined a priori using standardised definitions as per previous studies[7 37] (online supplementary appendix 3).

### Patient-reported outcomes

Participants will be asked to complete the preoperative breast reconstruction BREAST-Q questionnaire prior to surgery.

Postoperative pain scores will be collected by patient self-report using an electronic visual analogue scale at 24 hours, 1 week, 2 weeks and 3 months following surgery.

Patient satisfaction with breasts, physical well-being, psychosocial well-being, satisfaction with implants and patient-reported assessment of animation will be assessed at 3 and 18 months using the validated BREAST-Q questionnaire.[38] A reminder will be sent after 1 month if no response is received. Patients will be encouraged to complete online patient-reported outcome measures (PROMs) but paper versions will be made available to patients unable or unwilling to use electronic versions.

### Shared learning and intervention stability

Best practice and opportunities for shared learning among participating surgeons will occur at the following points in the study.

i. Study entry.
ii. As a result of procedure modifications or complications.

Surgeons electing to participate in the study will be sent a surgeon information sheet outlining the proposed shared learning study and what participation may involve. Surgeons will be able to recruit patients to Pre-BRA without contributing to shared learning but full participation will be encouraged by the steering group.

At each time point, a combination of free-text responses on standardised CRFs and focused semistructured qualitative interviews will be to explore group learning and intervention stability.

#### Study entry
##### Phase 1: surgeon questionnaire

Prior to commencing patient recruitment, participating surgeons will be asked to complete a surgeon questionnaire. The questionnaire (online supplementary

appendix 4) will explore techniques and products used; experience; approaches to concomitant interventions and patient selection criteria. The questionnaire will include consent to be contacted regarding a brief (20 min) semistructured qualitative interview exploring their practice in more detail.

### Phase 2: semistructured study entry interviews

#### Sampling and data collection
A purposive sample of surgeons consenting to interview will be selected based on techniques and products used; experience; geographical location and differing approaches to concomitant interventions and patient selection.

Interviews will be organised at a time convenient to the participating surgeons. All interviews will be conducted by telephone using a semistructured topic guide developed based on the literature and clinical expertise to explore issues surrounding current practice. All interviews will be audiorecorded and transcribed verbatim. Interview participants will be asked to give verbal consent to study participation and audiorecording prior to commencing the interview. The topic guide will be modified iteratively as the study progresses to allow new themes to emerge. Sampling, data collection and analysis will be conducted concurrently and iteratively until data saturation is achieved and no new themes emerge from the data. As a result of procedure modifications and learning from complications.

### Phase 1: case report forms
Surgeons will be asked to report (yes/no) whether (i) a modification to the planned procedure was made at the time of the operation and (ii) any learning (eg, changes to future surgical technique) occurred as a result of a complication at 3 months. If modifications or learning as a result of complications are reported, surgeons will be asked to provide details in a free-text response box on the electronic CRF and to indicate whether they would be willing to be contacted to participate in a brief telephone interview to explore this in more detail.

### Phase 2: semistructured surgeon modification/complication interviews

#### Sampling and data collection
Surgeons providing consent will be contacted to arrange a brief semistructured qualitative interview. Interviews will be conducted as per study entry interviews outlined above but will focus specifically on the modification/learning arising from the complication including details, rationale and outcome.

The feasibility of this approach is unknown so initially, it is planned that all surgeons reporting procedure modifications/learning from complications and providing consent will be interviewed. As the study progresses, depending on the number and content of the CRF reports, all consenting surgeons may continue to be interviewed or, it may be necessary to purposively sample consenting surgeons based on the details of the modification/learning reported in the CRF to explore new and emerging themes and allow the details of the modifications to be explored. Data collection will continue until no new learning or modifications emerge from the data.

### Data analysis and sample size
#### Prepectoral safety and effectiveness
##### Sample size
A single-arm two-stage design will be used to assess the safety of PPBR based on the primary outcome of implant loss rate at 3 months. It is expected that the implant loss rate will be 9% or lower based on findings from the recent iBRA study;[7] 15% or greater will be deemed unacceptably high.

A sample size of 310 patients would result in a two-sided 95.0% CI for a single proportion, assumed to be 0.10, with a width equal to 0.070. Allowing for a 10% loss to follow-up at 3 months at least 341 patients will be recruited to inform a future trial with implant loss as the primary outcome.

Based on data from the iBRA study, the number of prepectoral reconstructions performed at each site is likely to be small (4–40 per year).[7] We will therefore aim to recruit from between 20 and 30 sites over a period of 12 months.

### Statistical analysis
Simple summary statistics will be calculated to describe demographic, procedure, process and outcome data. Categorical data will be summarised by counts and percentages and continuous data by median, IQR and range.

We will establish the proportion and 95% CIs of patients experiencing implant loss and other key safety outcomes (readmission for complications, reoperation and infection) at 3 months and compare these with findings with those reported in the iBRA study.[7]

An exploratory risk factor analysis will be conducted to explore the impact of type of mesh and known risk factors for complications including smoking; BMI and radiotherapy if sufficient numbers of patients are recruited to the study.

### Planned interim analysis
A planned interim analysis will be performed when 155 patients have been recruited to the study and followed-up for a minimum of 3 months. The report will be reviewed by the study steering group and the results will be made available to the study sponsor.

If the overall implant loss rate at 3 months following surgery is 15% or more, the study will be stopped.

### Shared learning and intervention stability feasibility study
Analysis of the free-text responses will be an ongoing and iterative process commencing soon after the study begins and will inform sampling for the semistructured interview phase of the study. Text will be assigned codes and analysed

using the constant comparative technique of grounded theory.[39] A combination of thematic and content analysis will be used to identify common themes.[40 41] These will inform the topic guide for the interviews and the feedback provided to study participants.

Sampling, data collection and analysis will be undertaken concurrently and iteratively until no new themes emerge from the data.

### Dissemination of best practice and shared learning
The learning and modifications emerging from the CRFs and interviews will be consolidated and reviewed at regular intervals by the study team. Any common themes will be shared with participating surgeons using a combination of methods. These will include a 'tips and tricks' section on the study website; e-mail updates; study newsletters and potentially social media channels (eg, 'You Tube'). The acceptability and value of the different approaches to sharing learning will be discussed with individual surgeons at interview and the wider steering group. Metrics such as number of views/hits for online/ social media resources will be used to provide additional information regarding engagement if appropriate.

### Design of the main study
If PPBR is considered safe and data from the screening logs suggests sufficient numbers of PPBR are being performed at participating centres, a consensus meeting with key participating surgeons, steering group members, patients and methodologists will be held to agree the final design for a large-scale pragmatic RCT comparing prepectoral and subpectoral IBBR.

### DISSEMINATION
Results will be analysed and presented at national and international meetings and published in peer-reviewed journals. Findings will be fed back to the surgical community to promote engagement and recruitment to a future RCT.

**Author affiliations**
¹National Institute for Health Research Bristol Biomedical Research Centre, University Hospitals Bristol NHS Foundation Trust and University of Bristol, Bristol, UK
²Applied Statistics Group, University of the West of England, Bristol, UK
³Breast Unit, Royal Liverpool University Hospital, Liverpool, UK
⁴Bristol Breast Care Centre, North Bristol NHS Trust, Westbury on Trym, UK

**Collaborators** The Pre-BRA Feasibility Study Steering Group;Peter Barry; Simon Cawthorn; Matthew Gardiner; Gareth Irwin; Cliona Kirwan; Mairead McKenzie; Shireen McKenzie; Rachel O'Connell; Georgette Oni; Tim Rattay; Pankaj Roy; Joanna Skillman; Soni Soumian; Raghavan Vidya; Lisa Whisker; Samantha Williams.

**Contributors** SP and CH conceived the study. KLH, SP, NM and CH contributed to the study design. PW provided statistical support. SP, CH and KLH secured funding for the project. KLH wrote the paper, SP critically revised the manuscript. All authors read, critically revised and approved the manuscript before submission.

**Funding** This work was supported by a One Year Royal College of Surgeons Blond McIndoe Foundation Research Fellowship, the Association of Breast Surgery and the NIHR Biomedical Research Centre at University Hospitals Bristol NHS Foundation Trust and the University of Bristol. SP is an NIHR Clinician Scientist

(CS-2016-16-019). The views expressed are those of the authors and not necessarily those of the NIHR or the Department of Health and Social Care.

**Disclaimer** This article is part of a series proposed by the Countdown to 2030 for Women's, Children's and Adolescents' Health and the Partnership for Maternal, Newborn & Child Health (PMNCH) hosted by the WHO and commissioned by The BMJ, which peer reviewed, edited and made the decisions to publish. Open access fees are funded by the Bill and Melinda Gates Foundation and PMNCH.

**Competing interests** None declared.

**Patient consent for publication** Written informed consent will be obtained for study participation.

**Ethics approval** Full ethical approval has been obtained for the study (NRES OXFORD-B South Central Committee Ref:19/SC/0129. IRAS ID: 255421).

**Provenance and peer review** Not commissioned; externally peer reviewed.

**ORCID iD**
Shelley Potter http://orcid.org/0000-0002-6977-312X

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
