## [Reviewer comments · BMJ Open]

ARTICLE DETAILS

TITLE (PROVISIONAL)	The Pre-BRA (Pre-pectoral Breast Reconstruction Evaluation) Feasibility Study: Protocol for a mixed-methods IDEAL 2a/2b prospective cohort study to determine the safety and effectiveness of pre-pectoral implant-based breast reconstruction
AUTHORS	Harvey, Kate Louise; Mills, Nicola; White, Paul; Holcombe, Christopher; Potter, Shelley; Pre-BRA Study Steering Group, The

VERSION 1 – REVIEW

REVIEWER	Parto Forouhi Cambridge University Hospitals NHSFT United Kingdom
REVIEW RETURNED	22-Aug-2019

GENERAL COMMENTS	The authors should be congratulated on producing an outstanding piece of work. Their approach represents a new paradigm in evaluating surgical technology. This protocol is highly relevant to an evolving surgical practice and is a natural follow up to the iBRA study. The IDEAL framework chosen for the study, and the addition of qualitative assessment are appropriate and well applied. Please note the following proofing points: The abstract stated the study start date to be spring 2019: Is this an oversight or has the study already started. On page 11, line 31, page 51, line 48 the abbreviation PBR is used. Is this meant to be PPBR? I have the following comments regarding Inclusion and exclusion criteria: There should be a single set of exclusion criteria. The breakdown into absolute and relative exclusion criteria is problematic. This is a prospective study of an evolving practice and as such needs to be widely inclusive. With a relatively small sample size, inclusion of outliers of practice however may bias the results. What has been termed “relative exclusion criteria” represent what would usually be termed “relative contraindications”. The protocol could simply state that that “it is expected that the procedure would be offered according to ABS/ BAPRAS guidelines”. Offering PPBR in the presence of two or more relative contraindications could then be considered an outlier of practice and be an appropriate reason for exclusion from the study. The exclusion criterion 1 (Page 8, line 29): is also somewhat problematic and not necessary. It is generally accepted that poor flaps are a contraindication to implant based breast reconstruction. Flap quality is highly subjective part of the surgeon’s decision
--

	making process. Offering the procedure once such a judgement has been made about the flaps would be considered a medical mistake and not just a practice outlier. Regarding Secondary safety and effectiveness outcomes (page 12, lines 13 onwards and complications: (page 47 line 43) Consideration should be given to including persistent non infective breast erythema (“red breast syndrome).
REVIEWER	Andreas Karakatsanis Department for Surgical Sciences, Faculty for Medicine, Uppsala University, Sweden
REVIEW RETURNED	29-Oct-2019
GENERAL COMMENTS	This is a well designed protocol for a necessary study

VERSION 1 – AUTHOR RESPONSE

Reviewer: 1

1. The authors should be congratulated on producing an outstanding piece of work. Their approach represents a new paradigm in evaluating surgical technology. This protocol is highly relevant to an evolving surgical practice and is a natural follow up to the iBRA study. The IDEAL framework chosen for the study, and the addition of qualitative assessment are appropriate and well applied. We thank Reviewer 1 for their kind comment.

2. Please note the following proofing points: The abstract stated the study start date to be spring 2019: Is this an oversight or has the study already started.
Thank you for asking for clarification on this point. This is not a typographical error: the study opened to recruitment June 2019 and will recruit for 12 months.

3. On page 11, line 31, page 51, line 48 the abbreviation PBR is used. Is this meant to be PPBR?
This has been revised.

4. I have the following comments regarding Inclusion and exclusion criteria: There should be a single set of exclusion criteria. The breakdown into absolute and relative exclusion criteria is problematic. This is a prospective study of an evolving practice and as such needs to be widely inclusive. With a relatively small sample size, inclusion of outliers of practice however may bias the results. What has been termed “relative exclusion criteria” represent what would usually be termed “relative contraindications”. The protocol could simply state that that “it is expected that the procedure would be offered according to ABS/ BAPRAS guidelines”. Offering PPBR in the presence of two or more relative contraindications could then be considered an outlier of practice and be an appropriate reason for exclusion from the study.

We thank Reviewer 1 for their considered comment. The steering group discussed the inclusion and exclusion criteria for the study at length. We know from the iBRA study (Potter et al, Lancet Oncology 2019) that implant-based reconstruction in general is not performed according to ABS/BAPRAS guidelines and that a high proportion of procedures are performed in high-risk groups. Discussions with the steering group highlighted that surgeons with more experience of performing prepectoral reconstruction would offer the procedure to patients who would be considered high risk according to ABS/BAPRAS guidelines. If these groups were excluded, it would not stop these surgeons performing the procedure and we would lose this valuable opportunity to collect outcome data in higher risk patients. We have therefore elected to include all patients considered by their operating surgeon to be

suitable for prepectoral implant-based reconstruction in the current study (and to collect detailed information on risk factors) to reflect real-world practice.

5. The exclusion criterion 1 (Page 8, line 29): is also somewhat problematic and not necessary. It is generally accepted that poor flaps are a contraindication to implant based breast reconstruction. Flap quality is highly subjective part of the surgeon's decision making process. Offering the procedure once such a judgement has been made about the flaps would be considered a medical mistake and not just a practice outlier.

This is again an important point. It is recommended that patients undergoing prepectoral reconstruction have good soft tissue coverage with several authors recommending a 1cm pinch test in the upper chest at a preoperative assessment to be considered suitable for the technique. We have revised the exclusion criteria for clarity:

Absolute exclusion criteria, page 8 now reads

'i. Patients assessed as having insufficient soft tissue coverage for PPBR by the operating surgeon'

6. Regarding Secondary safety and effectiveness outcomes (page 12, lines 13 onwards and complications: (page 47 line 43). Consideration should be given to including persistent non infective breast erythema ("red breast syndrome).

Thank you for identifying this omission. We have added red breast syndrome to the list of complications in Appendix 3.

Reviewer: 2

7. This is a well designed protocol for a necessary study

We thank Reviewer 2 for their kind comments.